# Description of Two Fungal Endophytes Isolated from *Fragaria chiloensis* subsp. *chiloensis* f. *patagonica*: *Coniochaeta fragariicola* sp. nov. and a New Record of *Coniochaeta hansenii*

Carolina Campos-Quiroz [1], Jean Franco Castro [1,*], Cecilia Santelices [1], Jorge Carrasco-Fernández [1], Matías Guerra [1], Diego Cares-Gatica [1], Javiera Ortiz-Campos [1], Yocelyn Ocares [1], Lorena Barra-Bucarei [1] and Bart Theelen [2]

1   Instituto de Investigaciones Agropecuarias, INIA Quilamapu. Av. Vicente Méndez 515, Chillán 3800062, Ñuble, Chile
2   Westerdijk Fungal Biodiversity Institute, 3584 CT Utrecht, The Netherlands
*   Correspondence: jean.castro@inia.cl; Tel.: +56-42-2206786

**Abstract:** Prospection of the endosphere of the native plant *Fragaria chiloensis* subsp. *chiloensis* f. *patagonica* from the foothills of the Chilean Andes led to the isolation of two strains of the genus *Coniochaeta*. We addressed the taxonomic placement of these strains based on DNA sequencing data using the ITS and LSU genetic markers, morphological features, and biochemical traits. One of these strains was identified as *Coniochaeta hansenii*, for which the anamorph and teleomorph states were described. The second strain did not seem to match any of the currently described species of this genus; therefore, we propose the name *Coniochaeta fragariicola* sp. nov.

**Keywords:** biochemical tests; *Coniochaeta* taxonomy; plant endophytes

## 1. Introduction

The genus *Coniochaeta* (Sacc.) Cooke [1] (*Coniochaetaceae*, *Coniochaetales*, *Ascomycota*) is characterized by strongly melanized, cleistothecial or perithecial, usually setose ascomata with cylindrical, unitunicate asci, and one-celled ascospores with a germ slit. Anamorphs are predominantly phialidic and often produce intercalary conidiogenous cells with lateral necks, giving rise to one-celled conidia in slimy masses, with some members developing microcyclic conidiation [2,3]. According to Index Fungorum (www.indexfungorum.org, (accessed on 12 December 2022)), the genus includes 117 species, many of which have only been described based on their morphological characteristics, with no information concerning sequenced DNA regions for phylogenetic analysis. This group of fungi occurs in both terrestrial and aquatic environments and has been reported on bark, dung, lichens, soil, wood, polluted water, submerged plant material, and other substrates [4–6].

Previous studies have revealed that members of *Coniochaeta* may have either beneficial or harmful effects on plants. Many of its members occur as endophytes of phylogenetically diverse hosts, and some species produce antimicrobial secondary metabolites that aid in the protection against plant pathogenic species of *Colletotrichum*, *Verticillium*, and other genera [7,8]. On the other hand, certain *Coniochaeta* species have the ability to infect woody plants, such as apricot and peach [9], while others are lignicolous, humicolous, and coprohilus [10]. However, they are generally regarded as low-virulent, opportunistic pathogens that proliferate on wounded, senescent, or infected plant tissues [4]. Genome mining and transcriptome analyses have discovered that some members of *Coniochaeta* have exceptional lignocellulolytic machinery to deconstruct lignin [11]. Several *Coniochaeta* species, mostly known as anamorphs, have been implicated in superficial and deep-seated, life-threatening forms of phaeohyphomycosis in humans and dogs [12–14]. Other species are known for their teleomorphs, such as coprohilous *Coniochaeta* with poly-spored asci [15].

The taxonomic history of *Coniochaeta* is complex, and early phylogenetic studies found that, under its traditional circumscription, this genus was polyphyletic [16]. Nevertheless, in a molecular study conducted by García et al. [2], a redefinition of the genus was proposed, which included the synonymization of three morphologically similar genera, i.e., *Coniochaetidium*, *Ephemeroascus*, and *Poroconiochaeta* with *Coniochaeta*. Further taxonomic improvements were made with the transfer of several species originally described in *Rosellinia* and in the asexually typified genus *Lecythophora* to *Coniochaeta*, based on molecular and phenotypic data [12,17]. Despite all these advancements towards a more stable taxonomy of this genus, no DNA sequence data is currently available for many of its members and many species most likely remain undiscovered, especially in poorly studied regions.

During taxonomic studies on the microfungi associated with the native Chilean wild strawberry *Fragaria chiloensis* (*Rosaceae*, *Rosales*), two strains of the genus *Coniochaeta* were isolated. One of these strains did not seem to match any of the currently described species of this genus. In this paper, we addressed the taxonomic placement of these strains based on the ITS and LSU genetic markers, morphological features, and biochemical traits.

## 2. Materials and Methods

### 2.1. Sampling and Fungal Isolation

In 2020, 12 plants of *Fragaria chiloensis* subsp. *chiloensis* f. *patagonica* were collected from a forested area located in Pinto, Ñuble region, Chile (36.818367 S, 71.622217 W; altitude: 697 m.a.s.l.). All 12 plants were considered as one sample and were placed inside a sterile plastic bag and transported to the laboratory in a dark container. The sample was dissected into three sections: (i) roots, (ii) crown, and (iii) the phyllosphere containing the petioles and leaves. Each section was individually surface sterilized by submerging the tissues for 1 min in 70% ethanol, 3 min in 1.5% NaClO, and 1 min in 96% ethanol; finally, they were rinsed three times in sterile distilled water [18,19]. Nine tissues 5 mm in diameter were cut with a sterile scalpel and distributed on 24-well sterile microplates containing noble agar (NA). The plates were incubated at 25 °C for 2 wk and checked under a microscope every day for the emergence of endophytic fungi from the tissues. Upon emergence, fungal tips were removed using a sterile needle and inoculated on NA plates and incubated for 7 d at 25 °C. Two yeast-like strains were recovered and deposited in the Chilean Collection of Microbial Genetic Resources (CChRGM) of the Institute of Agricultural Research, Chile, and in the Culture Collection of Fungi and Yeasts (CBS) of the Westerdijk Fungal Biodiversity Institute, The Netherlands, under the codes RGM 3301 and RGM 3311, and CBS 149284 and CBS 149292, respectively.

### 2.2. DNA Sequence Analyses

Genomic DNA isolation was carried out with the Wizard Genomic DNA Purification Kit (Promega), following the manufacturer's instructions with the following modifications: 30–60 mg of fungal biomass from a 7 to 14 d old culture, grown in 10 mL of potato dextrose broth, was used as the initial material. The fungal biomass was placed in sterile 1.5 mL microcentrifuge tubes and, after adding a lysis solution, it was ground with a micropestle to disrupt the mycelial cell walls. Samples were then incubated in a water bath at 65 °C for 30 min, and 10 μL of 3 M sodium acetate was added for DNA precipitation. DNA concentration and purity were spectrophotometrically measured using a Take3 microplate reader (Epoch, Biotek Inc., Winooski, VT, USA) and the Gen5 software version 2.09 (Biotek Inc., Winooski, VT, USA).

The nuclear ribosomal internal transcribed spacer (ITS) region and a portion of the nuclear ribosomal large subunit RNA (LSU) gene were PCR amplified using the primers ITS1 (5′-TCCGTAGGTGAACCTGCGG-3′), ITS4 (5′-TCCTCCGCTTATTGATATGC-3′) [20] and LR0R (5′-ACCCGCTGAACTTAAGC-3′) [21], LR6 (5′-CGCCAGTTCTGCTTACC-3′) [22], respectively. The PCR program used for amplifying both genetic markers was the following: 95 °C for 5 min (initial denaturation), 35 cycles of 94 °C for 30 s (denaturation), 52 °C for 30 s (annealing), 72 °C for 1 min (extension), and 72 °C for 8 min (final extension). Each

amplification was verified in 1% agarose gels stained with GelRed (Maestrogen). Amplicons were sent to Macrogen (Republic of Korea) for Sanger sequencing with both forward and reverse primers. The ITS of *Coniochaeta hansenii* CBS 885.68 and *Coniochaeta leucoplaca* CBS 486.73 were amplified and sequenced at the Westerdijk Fungal Biodiversity Institute, Utrecht, The Netherlands, using the primers V9G (5′-TTACGTCCCTGCCCTTTGTA-3′) [23] and LS266 (5′-GCATTCCCAAACAACTCGACTC-3′) [24]. Nucleotide sequences were analyzed, trimmed, and assembled using Sequencher version 5.4.6 (Gene Codes, Ann Arbor, MI, USA). Basic Local Alignment Search Tool (BLAST) searches were carried out against the Nucleotide Collection (nr/nt) database using the ITS nucleotide sequences of strains RGM 3301 and RGM 3311. Nucleotide sequences for ITS and LSU of *Coniochaeta* reference strains were retrieved from previous studies [6,9,17,25–27] and downloaded from GenBank [28] (Table 1). Local BLAST searches were performed with SequenceServer [29], using a custom database. For phylogenetic purposes, multiple sequence alignments were sequentially conducted for a dataset of ITS and LSU nucleotide sequences using the T-Coffee [30] webserver with default parameters. Multiple sequence alignments were analyzed through the Transitive Consistency Score (TCS) webserver and manually edited [31]. A phylogenetic tree was estimated using the maximum likelihood tree-making algorithm on the IQ-TREE webserver with a 1000 SH-aLRT test and ultrafast bootstrap (UFboot) repetitions [32]. The best nucleotide substitution models for ITS1, 5.8S, ITS2, and LSU were TIM3ef+G, TIM3ef+I+G, K80+G, and TIM1+I+G, respectively; these were estimated using the Akaike Information Criterion (AIC) in jModelTest version 2.1.10 [33,34]. Each node of the phylogenetic tree displays the SH-aLRT [35] and UFBoot [36] support values, where SH-aLRT ≥ 80% and UFboot ≥ 95% indicate the reliability of the node. Phylogenetic data can be found at Dryad (https://datadryad.org/stash, accessed on 12 December 2022) [37]. All sequences obtained in this study were deposited in GenBank, and the accession numbers are listed in Table 1.

**Table 1.** GenBank accession number, collection number, host/source, and country of origin of *Coniochaeta* spp. used for the phylogenetic analyses.

| Fungal Species | Collection Number | Host/Source | Country | GenBank Accession | | Reference |
| | | | | ITS | LSU | |
|---|---|---|---|---|---|---|
| *Coniochaeta acaciae* | MFLUCC 17-2298[T] | *Acacia* sp. | Uzbekistan | MG062735 | MG062737 | [38] |
| *Coniochaeta africana* | CBS 120868[T] | *Prunus salicina* | South Africa | GQ154539 | GQ154601 | [9] |
| *Coniochaeta ambigua* | PAD S00027[T] | *Sambucus racemosa* | Italy | ITS1: MW626895, ITS2: MW626903 | - | [17] |
| *Coniochaeta angustispora* | CBS 144.70 | Trio compost | Netherlands | MH859528 | MH871308 | [27,39] |
| *Coniochaeta arenariae* | MFLUCC 18-0405[T] | *Ammophila arenaria* | United Kingdom | - | MN017893 | [5] |
| *Coniochaeta arxii* | CBS 192.89 | Soil | Japan | MH862164 | - | [17,27] |
| *Coniochaeta baysunika* | MFLUCC 17-0830[T] | *Rosa* sp. | Uzbekistan | MG828880 | MG828996 | [40] |
| *Coniochaeta boothii* | CBS 381.74[T] | Soil, mud | India | NR_159776 | AJ875226 | [2] |

**Table 1.** *Cont.*

| Fungal Species | Collection Number | Host/Source | Country | GenBank Accession | | Reference |
| | | | | ITS | LSU | |
|---|---|---|---|---|---|---|
| *Coniochaeta canina* | UTHSC 11-2460[T] R-4810[T] | Osteolytic lesion in a German shepherd dog | USA | JX481775 | JX481774 | [12,13] |
| *Coniochaeta cateniformis* | UTHSC 01-1644[T] | Canine bone marrow | USA | HE610331 | HE610329 | [41] |
| *Coniochaeta chordicola* | PAD S00030[T] | On a rope | Italy | ITS1: MW626898, ITS2: MW626905 | - | [17] |
| *Coniochaeta cephalothecoides* | L821 | Fruiting bodies of *Trametes* sp. | Tibet, China | KY064029 | KY064030 | [42] |
| *Coniochaeta cipronana* | CBS 144016[T] | Art lithograph | Costa Rica | NR_157478 | - | [43] |
| *Coniochaeta coluteae* | MFLUCC 17-2299[T] | *Colutea paulsenii* | Uzbekistan | MG137251 | MG137252 | [38] |
| *Coniochaeta cruciata* | FMR 7409 | Soil | Nigeria | - | AJ875222 | [2] |
| *Coniochaeta cymbiformispora* | NBRC 32199[T] | Swamp soil | Japan | LC146726 | LC146726 | Ban et al., unpublished |
| *Coniochaeta cypraeaespora* | CBS 365.93[T] | Dead grass from termite mound | South Africa | MH862420 | MH874073 | [27,44] |
| *Coniochaeta dakotensis* | PAD S00035[T] | *Crataegus* sp. | USA | ITS1: MW626902, ITS2: MW626910 | - | [17] |
| *Coniochaeta deborreae* | CBS 147215[T] | Soil | Belgium | MW883413 | MW883808 | [45] |
| *Coniochaeta decumbens* | CBS 153.42[T] | Strawberry fruit, *Fragaria* sp. | The Netherlands | NR_144912 | NG_067257 | [41] |
| *Coniochaeta dendrobiicola* | MCC 1811[T] | Roots of *Dendriobium lognicornu* | Nepal | MK225602 | MK225603 | [46] |
| *Coniochaeta discoidea* | CBS 158.80[T] | Paddy soil | Japan | NR_159779 | NG_064120 | [2] |
| *Coniochaeta discospora* | CBS 168.58[T] | Horse dung | Canada | MH857740 | MH869278 | [27] |
| *Coniochaeta elegans* | ARIZ-FF0093[T] | *Juniperus deppeana* | USA | MZ262389 | - | [26] |
| *Coniochaeta ellipsoidea* | CBS 137.68[T] | Soil | Japan | MH859091 | MH870804 | [27] |
| *Coniochaeta endophytica* | AEA 9094[T] | Endophyte of *Platycladus orientalis* | USA | EF420005 | EF420069 | [47] |
| *Coniochaeta euphorbiae* | CBS 139768[T] | Healthy plant of *Euphorbia polycaulis* | Iran | KP941076 | KP941075 | [25] |

**Table 1.** *Cont.*

| Fungal Species | Collection Number | Host/Source | Country | GenBank Accession | | Reference |
| | | | | ITS | LSU | |
|---|---|---|---|---|---|---|
| *Coniochaeta extramundana* | CBS 247.77[T] | Soil | USA | MH861057 | MH872828 | [27] |
| *Coniochaeta fasciculata* | CBS 205.38[T] | Butter | Switzerland | HE610336 | AF353598 | [41] |
| *Coniochaeta fibrosae* | CGMCC3.20304[T] | Medulla of the lichen *Candelaria fibrosa* | China | MW750760 | MW750758 | [6] |
| *Coniochaeta fodinicola* | CBS 136963[T] | Uranium mine raffinate | Australia | JQ904603 | KF857172 | [48] |
| **Coniochaeta fragariicola sp. nov.** | **RGM 3301[T] = CBS 149284[T]** | **Fragaria chiloensis subsp. chiloensis f. patagonica** | **Chile** | **OP962067 *** | **OP962069 *** | **This work** |
| *Coniochaeta geophila* | PAD S00031[T] | Sandy ground among mosses | Belgium | ITS2: MW626906 | - | [17] |
| *Coniochaeta gigantospora* | ILLS 60816[T] | Submerged wood of *Fraxinus excelsior* | France | JN684909 | JN684909 | [49] |
| *Coniochaeta hansenii* | CBS 885.68 | Dung of rabbit | The Netherlands | OP962072 * | AJ875223 | [2] |
| **Coniochaeta hansenii** | **RGM 3311 = CBS 149292** | **Fragaria chiloensis subsp. chiloensis f. patagonica** | **Chile** | **OP962068 *** | **OP962070 *** | **This work** |
| *Coniochaeta hoffmannii* | CBS 245.38[T] | Butter | Switzerland | HE610332 | AF353599 | [41,50] |
| *Coniochaeta iranica* | CBS 139767[T] | Healthy plant of *Euphorbia polycaulis* | Iran | KP941078 | KP941077 | [25] |
| *Coniochaeta krabiensis* | MFLU 16-1230[T] | Submerged marine wood | Thailand | - | MN017892 | [5] |
| *Coniochaeta leucoplaca* | CBS 486.73 | Agricultural soil | The Netherlands | OP962071 * | MH872465 | [27] |
| *Coniochaeta ligniaria* | CBS 110467 | *Picea abies* | Germany | - | AF353583 | [41,50] |
| *Coniochaeta ligniaria* | CBS 424.65 | Decaying wood in herbarium | USA | MH858650 | MH870292 | [51] |
| *Coniochaeta lignicola* | CBS 267.33[T] | Unknown | Sweden | NR_111520 | NG_067344 | [41,50] |
| *Coniochaeta lutea* | CBS 121445[T] | Necrotic wood of *Prunus salicina* | South Africa | GQ154541 | GQ154603 | [52] |
| *Coniochaeta luteorubra* | FMR 10721[T] | Leg wound | USA | HE610330 | HE610328 | [41] |
| *Coniochaeta luteoviridis* | CBS 206.38[T] | Butter | Switzerland | HE610333 | AF353603 | [41,50] |

**Table 1.** *Cont.*

| Fungal Species | Collection Number | Host/Source | Country | GenBank Accession | | Reference |
|---|---|---|---|---|---|---|
| | | | | ITS | LSU | |
| *Coniochaeta malacotricha* | F2105 | *Orthotomicus laricis* | Germany | - | AF353588 | [50] |
| *Coniochaeta marina* | MFLUCC 18-0408[T] | Sweden | Piece of driftwood retrieved from the sea | MK458764 | MK458765 | [53] |
| *Coniochaeta massiliensis* | PMML0158[T] | Human body (abscess on hand) | France | OM366153 | OM366268 | [54] |
| *Coniochaeta mongoliae* | CGMCC3.20250[T] | Medulla of the lichen *Ramalina sinensis* | China | MW077645 | MW077646 | [6] |
| *Coniochaeta montana* | ARIZ-SR0076[T] | *Juniperus deppeana* | USA | MZ262414 | - | [26] |
| *Coniochaeta mutabilis* | CBS 157.44[T] | River water | Germany | HE610334 | AF353604 | [41,50] |
| *Coniochaeta navarrae* | CBS 141016[T] | Bark of *Ulmus* sp. | Spain | KU762326 | KU762326 | [16] |
| *Coniochaeta nepalica* | NBRC 30584[T] | Soil | Nepal | LC146727 | LC146727 | Ban S. et al., unpublished data |
| *Coniochaeta nivea* | ARIZ-AK0926[T] | *Ophioparma ventosa* | USA | MZ262367 | - | [26] |
| *Coniochaeta ornata* | FMR 7415[T] | Soil | Russia | - | AJ875228 | [2] |
| *Coniochaeta ostrea* | CBS 507.70[T] | *Larrea* sp. twigs | USA | NR_159772 | NG_064080 | [2,27] |
| *Coniochaeta palaoa* | ARIZ-AEANC0604[T] | Endophytic in healthy *Hypnum* sp. | USA | MZ241149 | - | [52] |
| *Coniochaeta polymorpha* | CBS 132722[T] | Endotracheal aspirate of a preterm neonate | Kuwait | HE863327 | HE863327 | [12] |
| *Coniochaeta polysperma* | CBS 669.77[T] | Dung of hare | Japan | MH861109 | MH872868 | [27] |
| *Coniochaeta prunicola* | CBS 120875[T] | *Prunus armeniaca* | South Africa | GQ154540 | GQ154602 | [9] |
| *Coniochaeta pulveracea* | CBS 114628 | Rinsing machine, in soft drink factory | Turkey | MW883414 | GQ351560 | [45,55] |
| *Coniochaeta punctulata* | CBS 159.80 | River sludge | Japan | MH861254 | MH873024 | [27] |
| *Coniochaeta rhopalochaeta* | CBS 109872 | *Bulnesia retamas*, decorticated wood | Argentina | NR_172554 | GQ351561 | [56] |
| *Coniochaeta rosae* | TASM 6127[T] | Branches of *Rosa hissarica* | Uzbekistan | NR_157509 | NG_066204 | [40] |

**Table 1.** *Cont.*

| Fungal Species | Collection Number | Host/Source | Country | GenBank Accession ITS | LSU | Reference |
|---|---|---|---|---|---|---|
| *Coniochaeta savoryi* | CBS 725.74[T] | Wood of *Juniperus scopulorum* | United Kingdom | MH860890 | MH872627 | [2] |
| *Coniochaeta simbalensis* | NFCCI 4236[T] | Soil | India | NR_164024 | MG917738 | Rana and Singh, unpublished |
| *Coniochaeta sinensis* | CGMCC3.20306[T] | Medulla of the lichen *Ramalina sinensis* | China | MW422269 | MW422265 | [6] |
| *Coniochaeta sordaria* | CBS 492.73 | - | Germany | - | MH878380 | [6] |
| *Coniochaeta subcorticalis* | CBS 551.75 | *Pinus sylvestris* | Norway | MW883416 | AF353593 | [45,50] |
| *Coniochaeta taeniospora* | CBS 141014[T] | *Quercus petraea* | Austria | KU762324 | KU762324 | [16] |
| *Coniochaeta tetraspora* | CBS 139.68 | Soil | Japan | MH859093 | MH870806 | [27] |
| *Coniochaeta velutina* | CBS 981.68 | Waste stabilization pond | USA | MH859264 | MH870991 | [27] |
| *Coniochaeta velutinosa* | Co29[T] | Leaf of *Hordeum vulgare* | Iran | GU553327 | GU553330 | [57] |
| *Coniochaeta verticillata* | CBS 816.71[T] | Agricultural soil | The Netherlands | NR_159774 | AJ875232 | [2] |
| *Coniochaeta vineae* | KUMCC 17-0322[T] | Dead vine | China | NR_168225 | | [58] |
| *Chaetosphaeria innumera* | MR 1175 | *Fagus sylvatica* | Czech Republic | AF178551 | - | [59] |

[T] = ex-type isolates; * sequences obtained in this work; strains described in this work are in boldface; Abbreviations: AEM: A. Elizabeth Arnold; ARIZ: University of Arizona's Robert L. Gilbertson Mycological Herbarium, Tucson, AZ, USA; CBS: Centraalbureau voor Schimmelcultures, Utrecht, The Netherlands; CGMCC: China General Microbiological Culture Collection Center, Beijing, China; FMR: Facultad de Medicina, Reus, Tarragona, Spain; ILLS: Illinois Natural History Survey Fungarium, Champaign, IL, USA; KUMCC: Kunming Institute of Botany Culture Collection, Yunnan, China; MCC: Microbial Culture Collection (MCC) of the National Centre for Cell Science (NCCS), University of Pune Campus, Pune, India; MFLUCC: Mae Fah Luang University Culture Collection, Chiang Rai, Thailand; MFLU: Mae Fah Luang University, Chiang Rai, Thailand; MR: Martina Réblová; NBRC: NITE Biological Resource Center, Department of Biotechnology, National Institute of Technology and Evaluation, Kisarazu, Chiba, Japan; NFCCI: National Fungal Culture Collection of India, Agharkar Research Institute, Pune, India; PAD: Herbarium of the Padova Botanical Garden, Padua, Italy; PMML: Institut Hospitalo-Universitaire Méditerranée Infection, Marseille, France; RGM: Chilean Collection of Microbial Genetic Resources (CChRGM), Chillán, Chile; TASM: Tashkent Mycological Herbarium, Tashkent, Uzbekistan; UTHSC: Fungus Testing Laboratory, Department of Pathology, University of Texas Health Science Center at San Antonio, TX, USA.

*2.3. Macroscopic and Microscopic Characterization*

The macromorphological characteristics of the two isolates were described in three different media: potato dextrose agar (PDA, Difco[TM]), oatmeal agar (OA), and malt-extract agar (MEA, Difco[TM]). Five millimeter-sized discs were excised with a cork borer from the periphery of an actively growing fungal colony of each strain, previously grown on PDA at 20 °C for 7 d, and placed in the center of Petri dishes. The plates were incubated at 25 °C in the dark for 21 d.

The micromorphological characteristics of the strain RGM 3311 were described in OA, rabbit dung agar (RDA), potato carrot agar (PCA), and Leonian's agar. This strain was

incubated at 25 °C in the dark for a period of 60 to 75 d. In the case of RGM 3301, the micromorphological characteristics were evaluated in the following media: MEA, V8 agar, OA, corn flour agar, malt agar, Sabouraud (Difco™)—glucose agar, synthetic poor nutrient agar (SNA) with sterile filter paper, well-water agar (20%), modified Leonian's agar, PCA, RDA, and PDA after incubation at 20 and 25 °C in the dark at 20 °C for 12h/12h light and dark intervals, for up to 60 d. The rabbit dung agar was prepared as follows: 20 g of rabbit dung was sterilized at 121 °C for two 15 min sessions; the dung was then ground and resuspended in 1 L of distilled water and supplemented with 20 g of agar; and RDA was sterilized at 121 °C for 15 min. Other media were prepared according to Crous et al. [60].

Microscopic structures were observed using a Nikon Eclipse 80i microscope and the software NIS-Elements version 2.2 (Nikon Instruments Inc., Tokyo, Japan). Samples were placed in 60% lactic acid and stained with 5% cotton blue. Micromorphometric measurements were reported with a 95% confidence interval.

### 2.4. Phenotypic Characterization

The isolates were examined for phenotypic properties using GEN III microplates (Biolog Inc., Hayward, CA, USA) to assess their ability to oxidize a broad range of carbon sources and resist inhibitory agents [61,62]. Strains RGM 3301 and RGM 3311 were inoculated in PDA and incubated at 25 °C for 10 d for biomass production. The GEN III microplates were inoculated with a conidial suspension of RGM 3301 in the viscous inoculating fluid A (IF-A) at a cell density of 0.025 (600 nm), while RGM 3311 was inoculated in IF-A with a ground mycelium at a cell density of 0.070 (600 nm). The inoculated plates were incubated at 25 °C for 11 d in the dark. Data from microplates were analyzed in an Epoch microplate spectrophotometer (Biotek Inc., Winooski, VT, USA) at 550 nm.

### 2.5. Data and Image Processing

Microscopic measurements and data from phenotypic characterization were processed in Google Spreadsheet. Images were edited using GIMP version 2.10.32 and Inkscape version 1.1.

### 3. Results and Discussion

#### 3.1. Taxonomic Position of the Isolates

Both endophyte strains isolated from *Fragaria chiloensis* subsp. *chiloensis* f. *patagonica* were affiliated with the genus *Coniochaeta*, based on a maximum likelihood (ML) analysis of the concatenated datasets of ITS and LSU (Figure 1).

BLAST searches using the ITS sequence of RGM 3301 showed identities of 95.77% with *Coniochaeta* sp. DSM 101981 (GenBank accession: KX096678), *Sordariomycetes* sp. A45 (KX611033), and *Sordariomycetes* sp. ARIZ:SR0072 (KP991848) as the best hits. In turn, the best hits when using the ITS sequence of RGM 3311 were fungal sp. 2735 YZ-2011 (HM439584) with a 99.14% identity, followed by *Sordariomycetes* sp. A45 (KX611033) and *Coniochaeta ligniaria* B121 (KX090317), both with an identity of 91.01%.

A BLAST search using the individual ITS and LSU nucleotide sequences against a local database of *Coniochaeta* spp. revealed identical sequences between RGM 3311 and *C. hansenii* CBS 885.68 (ITS identities: 183/183 (100%), no gaps, LSU identities: 456/456 (100%), no gaps), while differences in these nucleotide sequence alignments were found when comparing with the second-best hits, *Coniochaeta montana* ARIZ-AZ0093 (ITS identities: 346/374 (90.1%), 4 gaps) and *Coniochaeta discospora* CBS 168.58[T] (LSU identities: 864/879 (98.3%), no gaps). Moreover, the strain RGM 3311 formed a single clade with *Coniochaeta hansenii* CBS 885.68 with a high branch support (92.8/100, SH-aLRT/UFBoot) and a similar branch length, suggesting that both strains represent one and the same species.

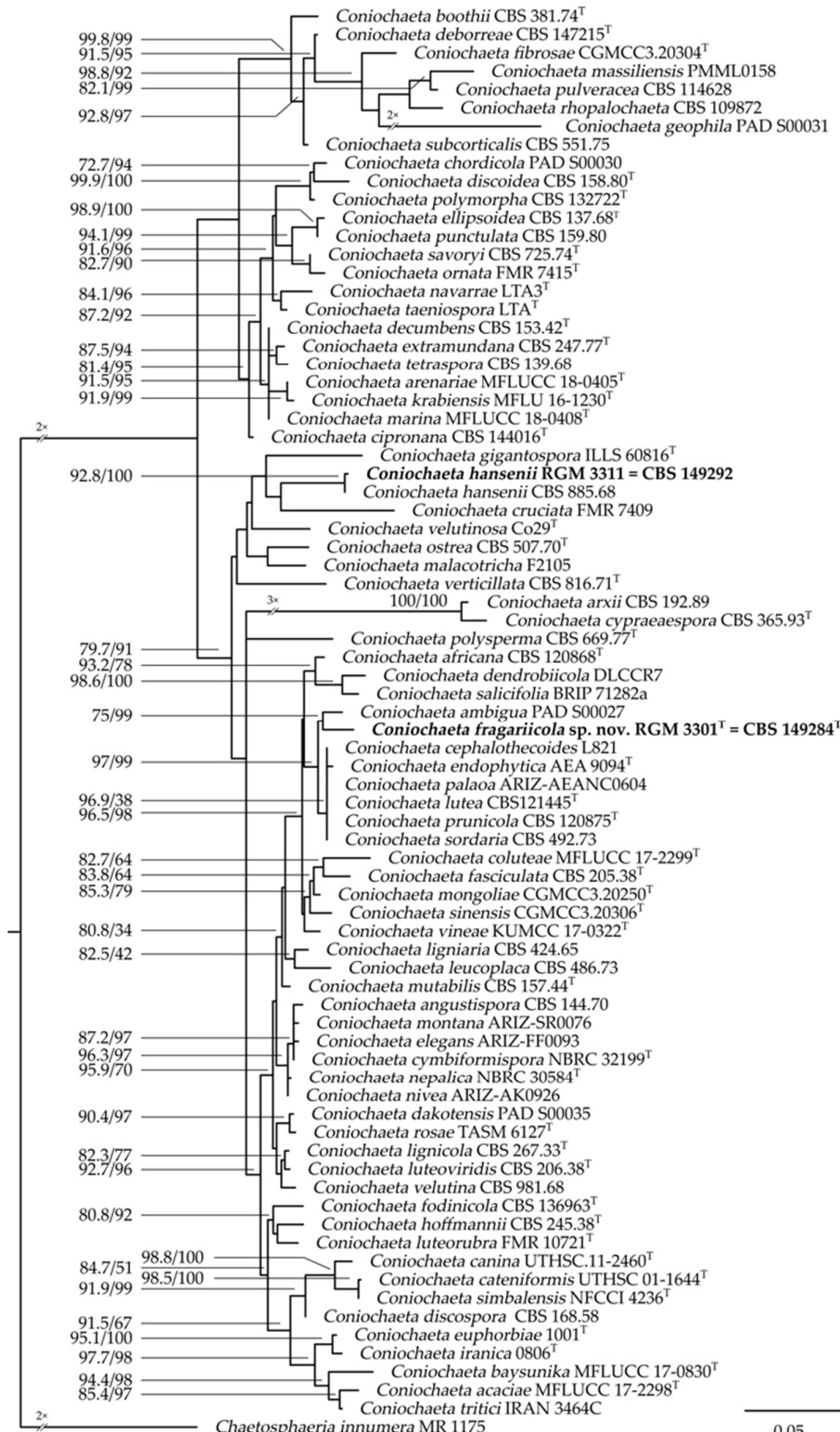

**Figure 1.** Maximum likelihood phylogenetic tree of *Coniochaeta* species inferred from W-IQ-TREE analysis of concatenated ITS and LSU data sets. Support values are represented by SH-aLRT/UFboot as numbers above the nodes. SH-aLRT ≥ 80% and UFboot ≥ 95% from 1000 bootstrap replicates are considered reliable branch support values. Support values at a node were not shown when both values were below the reliability threshold. Newly reported strains are shown in bold. *Ex-type* species are indicated with a "[T]". *Chaetosphaeria innumera* MR 1175 was used as an outgroup, and the tree was rooted to the outgroup.

*Coniochaeta cruciata* FMR 7409 and *Coniochaeta gigantospora* ILLS 60816[T], the closest neighbors to RGM 3311 in the ML tree, showed branch support values below the reliability threshold. In addition, similarities between the ITS and LSU nucleotide sequences of strain RGM 3311 with the respective genetic marker of *C. cruciata* FMR 7409 (no ITS sequence was available; LSU: 438/458 (96%), 2 gaps) and *C. gigantospora* ILLS 60816[T] (ITS identities: 495/548 (86.68%), 10 gaps; no LSU sequence was available) lead us to conclude that RGM 3311 is closely related to *C. hansenii* CBS 885.68 and distant to its closest neighbors. Therefore, it is proposed that strain RGM 3311 corresponds to *C. hansenii*.

On the other hand, the strain RGM 3301 was placed in the ML phylogenetic tree in a separate clade from RGM 3311. The strain RGM 3301 was grouped in a single clade with *Coniochaeta ambigua* PAD S00027[T], displaying a node support value that suggests they are related to one another (75/99, SH-aLRT/UFBoot). The nucleotide sequence alignment of the ITS1 and ITS2 of *C. ambigua* PAD S00027[T] compared to the respective sequences of *Coniochaeta* sp. RGM 3301 indicated low similarity values (ITS1 identities: 65/76 (86%), 3 gaps; ITS2 identities: 153/162 (94%), 6 gaps), suggesting that these two species are not closely associated.

According to a BLAST search against a *Coniochaeta* spp. local database, the best matches using the ITS nucleotide sequence of RGM 3301 were *Coniochaeta cephalothecoides* L821 (identities: 543/568 (94.54%), 5 gaps) and *Coniochaeta endophytica* AEA 9094[T] (identities: 536/562 (94.31%), 5 gaps), while when using the LSU nucleotide sequence the best matches were *C. endophytica* AEA 9094[T] (identities: 1006/1007 (99.21%), 2 gaps) and *Coniochaeta simbalensis* NFCCI 4236[T] (identities: 893/914 (97.7%), 21 gaps). The species *C. cephalothecoides* L821 and *C. endophytica* AEA 9094[T] were located in a separated clade, but next to that of the strain *Coniochaeta* sp. RGM 3301 (Figure 1). Altogether, these two sister clades showed a strong branch support value (97/99, SH-aLRT/UFBoot), indicating a close relationship among all these *Coniochaeta* species (Figure 1). It seems sensible to consider *Coniochaeta* sp. RGM 3301 is closely related to *C. ambigua* PAD S00027[T], *C. cephalothecoides* L821, *C. endophytica* AEA 9094[T], and *C. simbalensis* NFCCI 4236[T], but is, nonetheless, a separate species. Therefore, we propose the name *Coniochaeta fragariicola* sp. nov. for the strain RGM 3301[T].

### 3.2. Macromorphological Characterization of the Isolates

*Coniochaeta fragariicola* sp. nov. RGM 3301[T] and *C. hansenii* RGM 3311 colonies displayed yeast-like growth with entire margins, a circular shape, and sparse aerial mycelium in the three media tested: PDA, OA, and MEA (Figure 2 and Table 2). *Coniochaeta fragariicola* sp. nov. RGM 3301[T] showed different morphologies depending on the media tested. For instance, in PDA, the colony was rugose, with a pale-yellow pigmentation that changed to a brown-orange in the center after 15 d and yellow towards the margin after 21 d (Figure 2a). In OA and MEA, the colony had a glabrous appearance with a predominate pale orange color in the center and yellow-white towards the margin (Figure 2b,c). Moreover, in the MEA, the margin was more filamentous than in the other media (Figure 2c).

*Coniochaeta hansenii* RGM 3311 showed similar morphological characteristics in PDA (Figure 2d) and MEA (Figure 2f). For instance, colonies were shiny, with a moist appearance, radial grooves, and an undulating margin; they were orange in the center and yellow-gray towards the margin. In OA, the colony was drier, with an entire margin (Figure 2e). From the bottom, colonies showed similar characteristics to those observed in the respective media tested.

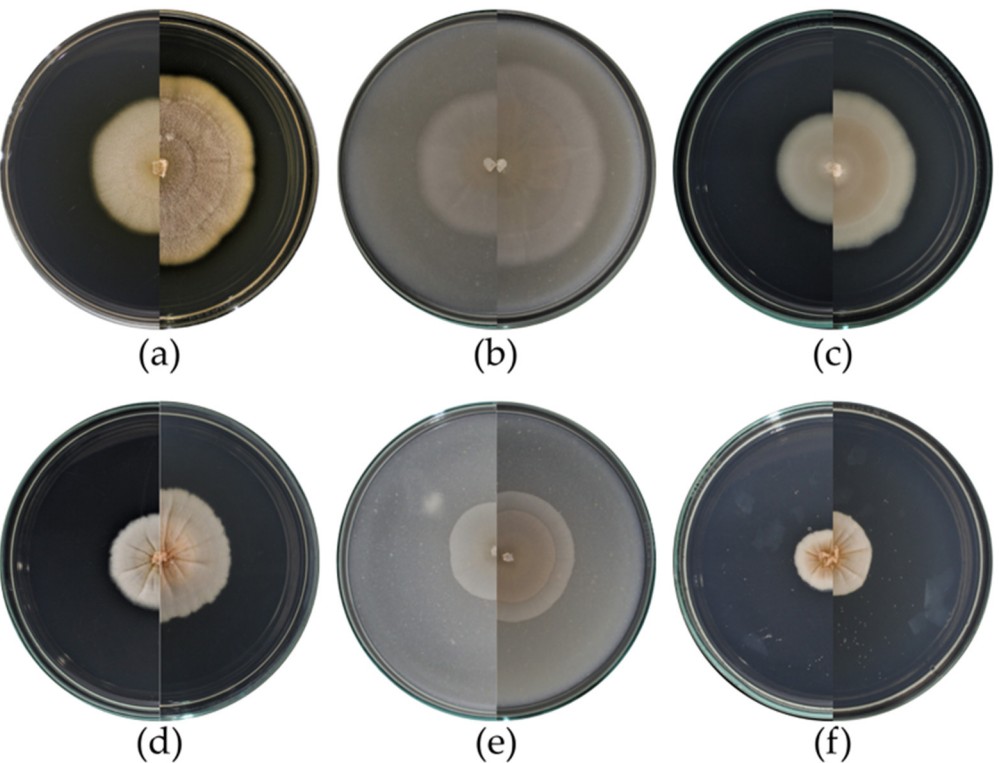

**Figure 2.** Cultural characteristics of *Coniochaeta* spp. isolated from *Fragaria chiloensis* subsp. *chiloensis* f. *patagonica*. *Coniochaeta fragariicola* sp. nov. RGM 3301[T] (top, (**a**–**c**)) and *Coniochaeta hansenii* RGM 3311 (bottom, (**d**–**f**)) growing on (**a**,**d**) potato dextrose agar (PDA), (**b**,**e**) oatmeal agar (OA), (**c**) malt-extract agar (MEA) at 25 °C for 15 d (half-left of each plate) and 21 d (half-right of each plate).

**Table 2.** Colony features of *Coniochaeta* spp. isolated from *Fragaria chiloensis* subsp. *chiloensis* f. *patagonica* in different media.

| Strain | Culture Medium [1] | | |
| --- | --- | --- | --- |
| | **PDA** | **OA** | **MEA** |
| *Coniochaeta fragariicola* RGM 3301 | Orange-brown in the center and faded-yellow towards the margin, flat colony with entire margin; from the bottom, the same characteristics. A growth diameter of 50 mm. Grows between 15 and 35 °C, optimal between 25 and 30 °C. | Pale orange-pink colony in the center and grayish-white towards the margin, flat with entire margin; from the bottom, the same characteristics. A growth diameter of 86 mm. | Bright pale orange-pink in the center and yellowish-gray towards the margin, flat colony with entire margin; from the bottom, the same color. A growth diameter of 45 mm. |
| *Coniochaeta hansenii* RGM 3311 | Orange with a slightly irregular margin, radial grooves from the center to the edge of the colony; from the bottom, the same coloration but paler. A growth diameter of 37 mm. Grows between 10 and 25 °C, optimal 25 °C. | Pale orange, flat with entire margin; from the bottom, the same characteristic. A growth diameter of 37 mm. | Pale orange, flat colony with entire margin, underside of the same coloration; from the bottom the same characteristics. A growth diameter of 23 mm. |

[1] Growth conditions: 25 °C for 21 d.

### 3.3. Micromorphological Characterization of the Isolates

Micromorphological characteristics of *Coniochaeta fragariicola* sp. nov. RGM 3301[T] = CBS 149284[T] were evaluated in several media. The anamorph of *C. fragariicola* sp. nov. RGM

3301[T] was obtained in PDA, OA, MEA, and PCA, whereas, for the teleomorph, only immature perithecia were produced in PCA and no further development was observed even though they were tested in all of the aforementioned media and conditions.

In PDA, hyphae were narrow, hyaline, multi-guttulate, and non-septate, with smooth walls, and frequently found in hyphal whorls (1.01–) 1.68–2.08 (–3.01) μm wide (Figure 3a,b). The conidiogenous cell was hyaline, smooth-walled, mostly ampulliform to lageniform, occasionally globose to subglobose, narrower at the base and wider in the middle, with a curved apex, (3.64–) 5.02–6.03 (–9.58) × (2.06–) 2.56–2.76 (–3.32) μm (Figure 3c–f), while in MEA and PCA the conidiogenous cell was subglobose to cylindrical and the apex was curved and also acute, with or without the formation of a collarette, showing microcyclic conidiation in MEA, after 8 d, at 25 °C (Figure 3g). Chlamydospores were hyaline, globose to subglobose, and they were produced in chains or groups of (2.69–) 3.94–4.47 (–5.57) μm, and were similar across all tested media (Figure 3h). Conidia were hyaline, smooth, and non-septate; allantoid demonstrated rounded ends and were generally bi-guttulate of (3.64–) 5.03–5.99 (9.58) μm × <1 μm (Figure 3i,j). Conidia with <1 μm were also observed in OA.

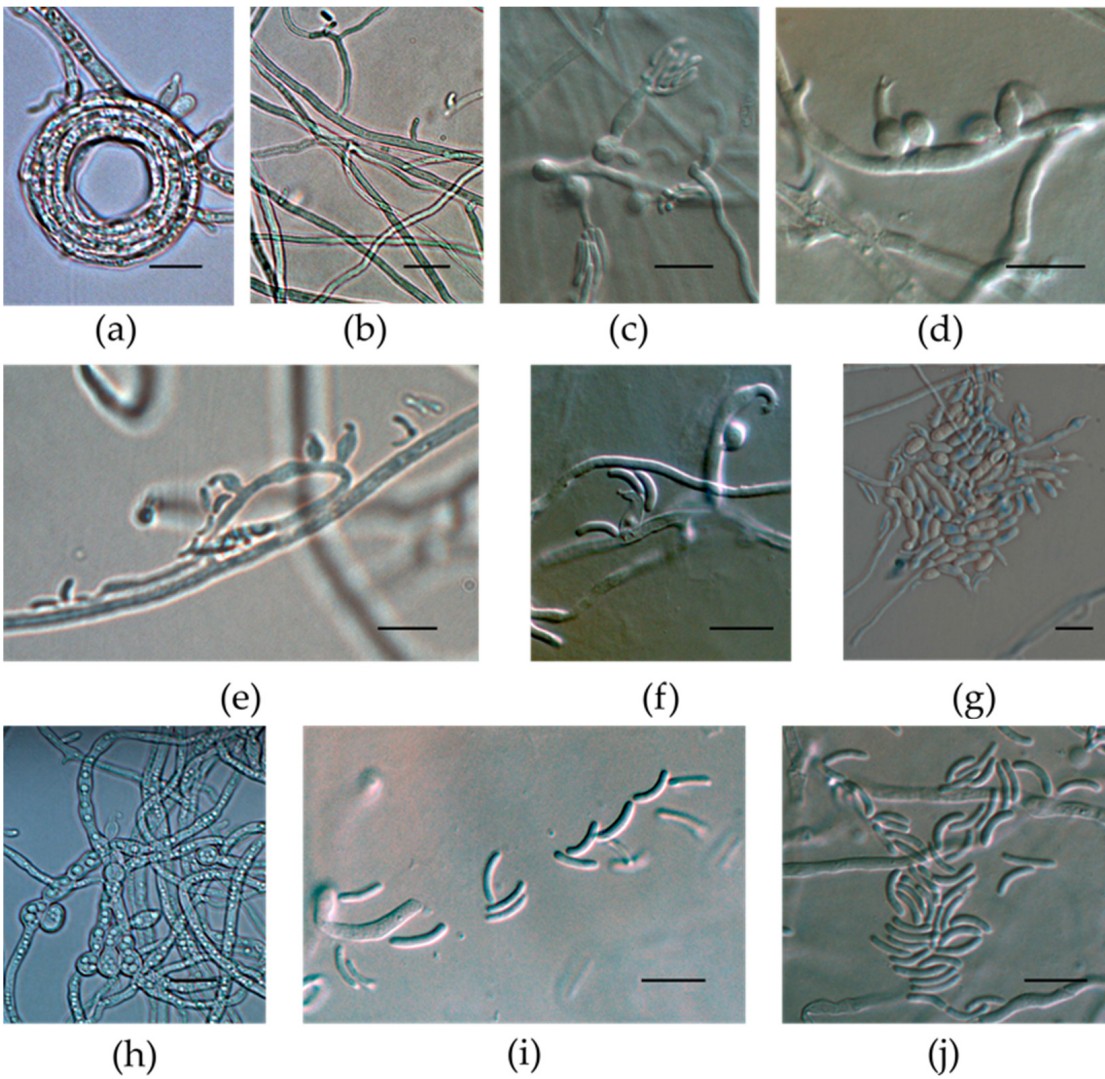

**Figure 3.** *Coniochaeta fragariicola* sp. nov. RGM 3301[T]. Hyphae (**a,b**), conidiogenous cell (**c–f**); collarette observed in (**d**), microcyclic conidiation (**g**), chlamydospores (**h**), and conidia (**i,j**). The structures were recorded in potato dextrose agar, except for (**g**), which was recorded in MEA. The bar scale was 10 μm.

Comparisons of *C. fragariicola* sp. nov. RGM 3301[T] were carried out with reference strains whose anamorph had been previously described, including the conidiogenous cell, conidia, microcyclic conidiation, and in some cases, the presence or absence of chlamydospores. Only the teleomorph of *Coniochaeta ambigua* PAD S00027[T] had been described; therefore, it was not possible to compare it with other strains. The conidiogenous cell of *C. endophytica* AEA 9094[T] [52] was characterized by reduced cylindrical protrusions from the hyphae or as discrete phialides; phialidic conidiogenous cells ampulliform were (5.2–10.3 × 2.3–2.9 μm wide), similar to *C. fragariicola* sp. nov. RGM 3301[T]. However, the conidia of *C. endophytica* AEA 9094[T] differed from those recorded for *C. fragariicola* sp. nov. RGM 3301[T], the former being shorter and wider in diameter ((2.5–) 3.1–3.4 (–4.4) μm × (1.3) 1.6–1.8 (2.4) μm) and ellipsoidal to fusiform in shape, while the conidia of *C. fragariicola* sp. nov. RGM 3301[T] were allantoid. *Coniochaeta endophytica*, unlike *C. fragariicola* sp. nov. RGM 3301[T], did not develop chlamydospores or microcyclic conidiation.

*Coniochaeta cephalotheicoides* L821 developed conidiogenous cells similar to those observed in RGM 3301[T], in that they were phialidic, terminal or lateral, borne on branched hyphae, hyaline, mostly ampulliform and sometimes ovoid or cylindrical (5–15 μm × 2.5–3.5 μm), with a distinct collarette. However, the conidia described for *C. cephalotheicoides* L821 were smaller, wider (2.5–5 μm × 1–2 μm), and more ellipsoidal than those observed in *C. fragariicola* sp. nov. RGM 3301[T] [63]. Additionally, the formation of mucous heads in the conidia of *C. cephalotheicoides* L821 was not observed in *C. fragariicola* sp. nov. RGM 3301[T]. *Coniochaeta palaoa* ARIZ-AEANC0604[T] and *C. fragariicola* sp. nov. RGM 3301[T] presented similarities in their phialide shape, but the conidial size was shorter and wider (3.6–4.7 μm × 1.5–1.9 μm) than the former [52]. In addition, *C. palaoa* ARIZ-AEANC0604[T] did not produce chlamydospores, unlike *C. fragariicola* sp. nov. RGM 3301[T] which did produce chlamydospores. Similarly, *Coniochaeta lutea* CBS 121445[T] developed conidia that were shorter and wider ((3.1) 3.7–4.1 (5.4) μm × (1.0) 1.3–1.6 (1.8) μm) than *C. fragariicola* sp. nov. RGM 3301[T]. The shape of the conidia in *C. lutea* CBS 121445[T] was bacilliform to allantoid [52], whereas in *C. fragariicola* sp. nov. RGM 3301[T] the conidia were allantoid. *Coniochaeta prunicola* displayed a similar conidia size ((2.5–) 3.5–6 (–8) μm × 1–2 (–3.0) μm) and shape, but microcyclic conidiation was not observed. *Coniochaeta sordaria*, another species close to *C. fragariicola* sp. nov. RGM 3301[T] in the phylogenetic tree, did not develop any anamorph stage; therefore, it was not possible to carry out morphological comparisons [64].

The conidiogenous cells of *Coniochaeta nivea* were usually originated from cylindrical protuberances of densely packed hyphae and only occasionally from phialides, which were unusual in *C. fragariicola* sp. nov. RGM 3301[T] [26]. Similarly, the conidiogenous cells in *Coniochaeta elegans* were mostly reduced to cylindrical protrusions from the hyphae, with collarettes, a condition that was not common in *C. fragariicola* sp. nov. RGM 3301[T], which presented mostly ampulliform conidiogenous cells with curved apices, regardless of whether or not they formed a collarette. Conidiation in *C. fragariicola* sp. nov. RGM 3301[T] occurs through the conidiogenous cell and by microcyclic conidiation, whereas conidiation in *Coniochaeta elegans*, as well as in *C. nivea*, occurs only through the conidiogenous cell [26]. Moreover, the conidia in *C. elegans* and *C. nivea* were shorter and wider when compared with *C. fragariicola* sp. nov. RGM 3301[T] (3.0–4.5 × 1.1–1.8 μm in *C. elegans*; 3.7–4.1 × 1.3–1.6 μm in *C. nivea*).

The conidia shape and length of *Coniochaeta angustispora* CBS 144.70 and *Coniochaeta dumosa* BF 42889 were similar to those recorded in the strain RGM 3301, but wider in *C. angustispora* and *C. dumosa* BF 42889 (1.25–1.75 μm and 1.5–2.5 μm, respectively); also, *C. angustispora* CBS 144.70 and *C. dumosa* BF 42889 displayed a slimy mass of conidia, which was not seen in *C. fragariicola* sp. nov. RGM 3301[T] [39,63]. The conidia of *C. fragariicola* sp. nov. RGM 3301[T] differed from those reported for *Coniochaeta coluteae* MFLUCC 17–2299[T], which were oblong, ellipsoidal to allantoid, and slightly shorter ((3.6) 4.5 ± 0.5 (6.3) μm) and wider ((0.7) 1.6 ± 0.3 (2.4) μm)); additionally, the conidiogenous cell was similar in morphology, but shorter ((3.3) 4.3 ± 0.5 (5.2) μm × (2.1) 3.1 ± 0.4 (3.8) μm) than that of *C. fragariicola*

sp. nov. RGM 3301[T] [38]. Likewise, the conidia of *Coniochaeta dendrobiicola* MCC 1811[T] were mostly cylindrical to allantoid and non-guttulate, unlike those of *C. fragariicola* sp. nov. RGM 3301[T], which were more homogeneous in size and shape (allantoid and guttulate). The sizes of the conidia of *C. dendrobiicola* MCC 1811[T] were significantly longer and wider (4.35–11.28 μm × 1.2–2.3 μm) than those observed in *C. fragariicola* sp. nov. RGM 3301[T] [46], while the conidia of *Coniochaeta montana* ARIZ-SR0076[T] were smaller and wider (3.4–4.5 μm × 1.9–2.1 μm) than those of *C. fragariicola* sp. nov. RGM 3301[T]. *C. montana* ARIZ-SR0076[T] did not present microcyclic conidiation or chlamydospores [26], as observed in *C. fragariicola* sp. nov. RGM 3301[T].

To the best of our knowledge, *C. hansenii* RGM 3311 = CBS 149292 corresponds to the first report of this species in Chile. In OA, *C. hansenii* RGM 3311 displayed pyriform, black ascomata with thick setae at the apex and finer setae (bristly hair) on the rest of the perithecium with a size of (195.46–) 348.41–412.03 (–618.55) μm × (193.52–) 270.37–311.07 (–387.51) μm; one ostiole was observed (Figure 4a,b). Setae were dark brown and straight, with pointed ends of (48.08–) 67.88–79.00 (–109.98) μm × (4.00–) 5.4–6.04 (–7.51) μm. Asci were clavate and slightly flattened at the apex, with the presence of an apical ring, enlarged subapical part, sometimes ascus with bifurcate apex and fusiform inflated, (85.96–) 105.42–116.98 (–142.11) μm × (13.47–) 16.88–18.27 (–21.92) μm (Figure 4c–f). Ascospores, nearly 128 in number, were brown, concave, smooth, not septate, discoidal, and (6.25–) 7.16–7.68 (–9.82) μm long × (5.43–) 5.85–6.07 (6.59) μm wide × (1.67–) 2.19–2.45 (–3.02) μm thick; they were ellipsoidal when viewed from the side and reddish-brown when old (Figure 4g,h). Hyphae hyaline were septate, smooth-walled, and (1.11–) 1.71–2.155 (–3.00) μm wide (Figure 4i). Chlamydospore hyaline were globose, clavate, doliiform, multiguttulate, and (5.37–) 9.19–10.59 (–14.1) μm × (5.69–) 7.42–8.30 (–10.28) μm (Figure 4j,k). The conidia were only observed in Leonian's agar and were sparse, hyaline, non-septate, cylindrical to ellipsoidal, occasionally fusiform, occasionally guttulate, and (4.1–) 4.79–5.23 (–6.67) μm × (0.93–) 1.37–1.52 (–2.31) μm wide (Figure 4l,m). Microcyclic conidiation was also observed.

The teleomorph of *C. hansenii* RGM 3311 was observed in OA, PDA, and RDA, while the anamorph was scarcely produced in Leonian's agar. In accordance with the molecular analysis, *Coniochaeta hansenii* RGM 3311 shared similar micromorphometric characteristics with *C. hansenii* (≡ *Sordaria hansenii* Oudem. 1882), as described by Oudemans [65]. The perithecia in this description were subglobose, measuring 350 μm in diameter, and featuring a neck with a short-conical shape, similar to those observed in *C. hansenii* RGM 3311. The setae in the original description were 23 μm long, which were about three-fold shorter than those observed in *C. hansenii* RGM 3311, though both terminated in bristly hairs. A report of *C. hansenii* in Italy [15] also corroborates the characteristics of the perithecium of *C. hansenii* RGM 3311, where both were brown to dark brown to black, pyriform, setose above the perithecium; however, it was bigger in the report from Italy (480–680 μm × 350–430 μm) [15].

The asci in *C. hansenii* RGM 3311 were shorter than those reported in both of the original descriptions and in the Italian strain (150 μm and 170–230 μm, respectively), though they shared the same shape overall [15,65]. In turn, the ascus width in RGM 3311 was similar to the original description (12 μm), but smaller when compared to the Italian report (20–27 μm). The key of *Coniochaeta* species with poly-spored asci, developed by Doveri [15] indicates that the number of spores per ascus is a distinctive feature for *C. hansenii* species. According to this key, the asci of *C. hansenii* species are clavate, rarely cylindric-clavate, measuring 125–230 μm × 17–32 μm, and containing 64 to 128 spores of a discoidal shape, measuring 6–10 μm × 4–9 μm × 3.5–7 μm. This general description matched that described here for *C. hansenii* RGM 3311. In addition, the Italian *C. hansenii* strain and *Coniochaeta polyspora* [66] contained possibly 128 spores per ascus, similar to *C. hansenii* RGM 3311, whereas, other *Coniochaeta* species, such as *Coniochaeta philocoproides* [67] and *Coniochaeta polymegasperma* [68] had 32-spored ascus, and *Coniochaeta polysperma* [69] 512-

spored ascus. Most *Coniochaeta* species whose teleomorph had been reported contained 8 spores per ascus [15].

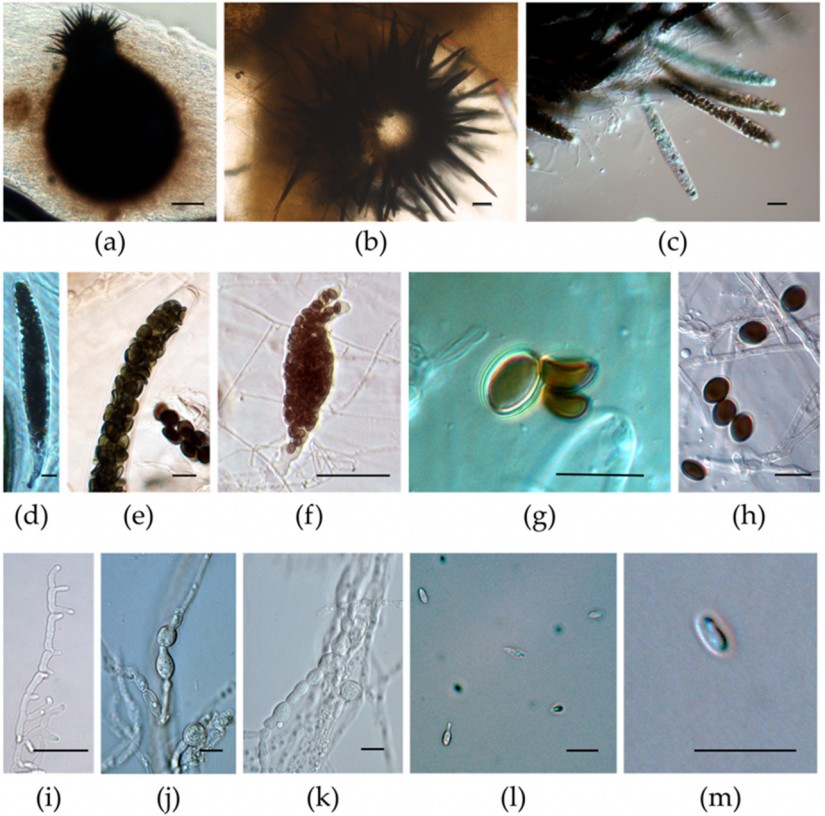

**Figure 4.** *Coniochaeta hansenii* RGM 3311. Perithecium, on a bar scale of 100 μm (**a**), perithecium ostiole, on a bar scale of 20 μm (**b**), asci, on a bar scale of 20 μm (**c**), individual ascus (**d**–**f**), ascus with bifurcate apex (**f**), bar scales: 10 μm (**d**), 20 μm (**e**), 50 μm (**f**); ascospores, bar scale of 10 μm (**g**,**h**); hyphae, bar scale of 10 μm (**i**); chlamydospores, bar scale of 10 μm (**j**,**k**); conidia (**l**,**m**). The structures were recorded in oatmeal agar, except for (**f**) which was observed in rabbit dung agar, (**i**–**k**) potato dextrose agar, and (**l**,**m**) Leonian's agar. The structures were recorded in oatmeal agar.

### 3.4. Biochemical Features of the Isolates

Both of the isolates *C. fragariicola* sp. nov. RGM 3301[T] and *C. hansenii* RGM 3311 oxidized the monosaccharides D-fructose-6-PO$_4$, L-fucose, α-D-glucose, D-mannose, methyl pyruvate, β-methyl-D-glucoside, and L-rhamnose, as well as the disaccharides D-cellobiose, gentiobiose, β-D-lactose, D-maltose, D-melibiose, and D-trehalose, sugars that were also used by *Coniochaeta massiliensis* [54]. Sucrose and stachyose oxidation were negative in *C. massiliensis*, while *C. fragariicola* sp. nov. RGM 3301[T] and *C. hansenii* RGM 3311 oxidized these disaccharides. Both strains oxidized polymers such as dextrin and pectin, as well as the amino-sugar *N*-acetyl-D-glucosamin; however, only the strain *Coniochaeta hansenii* RGM 3311 utilized *N*-acetyl-D-galactosamine, *N*-acetyl-β-D-mannosamine, and *N*-acetyl neuraminic acid. In relation to these sugars, enzymes such as glycoside hydrolases, glycosyl transferases, and polysaccharide lyases have been identified in *Coniochaeta* spp. [11,70]. Both isolates also oxidized sugar alcohols such as D-arabitol, glycerol, and D-salicin, while D-sorbitol, D-mannitol, and myo-inositol were only oxidized by *C. fragariicola* sp. nov. RGM 3301[T], which suggests that this strain presented an active mannitol cycle (positive for D-mannitol, D-fructose-6-PO$_4$, and D-fructose) [71]. Our isolates utilized organic acids acetic acid, γ-amino-butyric acid, citric acid, formic acid, L-galacturonic acid lactone, D-galacturonic acid, D-gluconic acid, D-glucuronic acid, α-hydroxy-butyric acid, β-hydroxy-D,L-butyric acid, α-keto-butyric acid, α-keto-glutaric acid, L-lactic acid, D-lactic acid methyl ester, D-malic acid, L-malic acid, mucic acid, propionic acid, and quinic acid.

*Coniochaeta fragariicola* sp. nov. RGM 3301$^T$ and *C. hansenii* RGM 3311 oxidized amino acids such as L-alanine, L-glutamic acid, glycyl-L-proline, and L-pyroglutamic, while only *C. hansenii* RGM 3311 used D-amino acids such as D-aspartic and D-serine. Both strains were inhibited by 4% and 8% NaCl, guanidine HCl, niaproof 4, and lithium chloride, but not by 1% NaCl. These strains were not inhibited by antibiotics such as aztreonam, fusidic acid, lincomycin, minocycline, nalidixic acid, rifamycin SV, troleandomycin, and vancomycin, and they grew in the presence of potassium tellurite, tetrazolium salts, D-serine, sodium bromate, and 1% sodium lactate; only sodium butyrate inhibited *C. hansenii* RGM 3311, while it did not inhibit *C. fragariicola* sp. nov. RGM 3301$^T$.

### 3.5. Taxonomy of the Two Endophytes of Fragaria chiloensis subsp. chiloensis f. patagonica

*Coniochaeta fragariicola* Campos-Quiroz and Castro *sp. nov.*—Mycobank 846930; Figure 3.

Etymology. "*fragari-*", reflects the host from which the fungus was isolated, *Fragaria chiloensis* subsp. *chiloensis* f. *patagonica*. "*-icola*", dweller.

In PDA. *Hyphae* hyaline, multi-guttulate, smooth-walled, sometimes helicoid in shape, non-septate, narrow, and (1.01–) 1.68–2.08 (–3.01) μm wide. *Conidiogenous cell* hyaline, smooth, mostly ampulliform to lageniform, occasionally globose to subglobose, narrower at the base and wider in the middle, with a curved apex, and (3.64–) 5.02–6.03 (–9.58) μm × (2.06–) 2.56–2.76 (–3.32) μm. On the contrary, in MEA and PC, the conidiogenous cell was subglobose to cylindrical, with a curved apex, sometimes a collarette was also formed. *Microcyclic conidiation* in MEA, after 8 d, at 25 °C. *Chlamydospores* hyaline, globose to subglobose, and produced in chains or groups, (2.69–) 3.94–4.47 (–5.57) μm. *Conidia* hyaline, smooth, non-septate, allantoid, with rounded ends that were generally bi-guttulate, and (3.64–) 5.03–5.99 (9.58) μm × <1 μm, L/W ratio = 4.9 ± 1.3.

Material examined: CHILE, Pinto, Ñuble, 36.818367 S, 71.622217 W; altitude: 697 m.a.s.l.; endophyte of *Fragaria chiloensis* subsp. *chiloensis* f. *patagonica*, 13 July 2020. Collected by J.F. Castro and M. Guerra, isolated by Cecilia Santelices. The holotype was deposited in the Chilean Collection of Microbial Genetic Resources (CChRGM) under the code RGM 3301 (the isotype was deposited in the CBS-KNAW culture collection under the code CBS 149284).

Notes: *Coniochaeta fragariicola* sp. nov. grew at a pH of 5 and 6 and in the presence of 1% NaCl. *Coniochaeta fragariicola* sp. nov. oxidized acetic acid, *N*-acetyl-D-glucosamine, L-alanine, γ-amino-butyric acid, D-arabitol, L-arginine, L-aspartic acid, L-butyric acid, citric acid, D-cellobiose, dextrin, formic acid, D-fructose, D-fructose-6-PO$_4$ (weakly), L-glucose, L-galactonic acid lactone, D-galactose, D-galacturonic acid, gentiobiose, D-gluconic acid, α-D-glucose, D-glucuronic acid, L-glutamic acid, glycerol, glycyl-L-proline, α-hydroxybutyric acid, β-hydroxy-D,L-butyric acid, α-keto-butyric acid, α-keto-glutaric acid, L-lactic acid, D-lactic acid methyl ester, β-D-lactose, D-malic acid, L-malic acid, D-maltose, D-mannitol (weakly), D-mannose, D-melibiose, β-methyl- D-glucoside, methyl pyruvate, mucic acid (weakly), myo-inositol, L-pyroglutamic acid, pectin, propionic acid, quinic acid, D-raffinose (weakly), L-rhamnose, D-salicin, L-serine, D-sorbitol, D-trehalose, D-turanose, and Tween 40, though it did not oxidize sucrose, stachyose, *N*-acetyl-β-D-mannosamine, *N*-acetyl-D-galactosamine, *N*-acetyl neuraminic acid, 3-methyl glucose, D-fucose, inosine, D-glucose-6-PO$_4$, D-aspartic acid, gelatin, L-histidine, glucuronamide, D-saccharic acid, *p*-hydroxy-phenylacetic acid, bromo-succinic acid, and acetoacetic acid. It was sensitive to 4% NaCl, 8% NaCl, guanidine HCl, niaproof 4, and lithium chloride, but not to aztreonam, fusidic acid, lincomycin, minocycline, nalidixic acid, rifamycin SV, troleandomycin, vancomycin, D-serine, sodium bromate, sodium butyrate, 1% sodium lactate, potassium tellurite, tetrazolium blue, or tetrazolium violet.

*Coniochaeta hansenii* (Oudem.) Cain, Studies of Coprophilous Spaeriales in Ontario: 63 (1934) (Figure 4).

≡ *Sordaria hansenii* Oudem., Hedwigia, 21: 123 (1882) (basionym).

≡ *Philocopra hansenii* (Oudem.) Oudem., Hedwigia, 21: 160 (1882).

In OA. *Ascomata* pyriform, black, with thick setae at the apex and finer setae (bristly hairs) on the rest of the perithecium; they were (195.46–) 348.41–412.03 (–618.55) μm × (193.52–) 270.37–311.07 (–387.51) μm, and one ostiole was observed. *Setae* dark brown, straight, with pointed ends, and (48.08–) 67.88–79.00 (–109.98) μm × (4.00–) 5.4–6.04 (–7.51) μm. *Asci* clavate, and slightly flattened at the apex, with the presence of an apical ring, enlarged subapical part, sometimes ascus showed a bifurcate apex and fusiform, and were (85.96–) 105.42–116.98 (–142.11) μm × (13.47–) 16.88–18.27 (–21.92) μm. *Ascospores* totaled nearly 128, and were brown, concave, smooth, not septate, discoidal from the top, and (6.25–) 7.16–7.68 (–9.82) μm long × (5.43–) 5.85–6.07 (6.59) μm wide × (1.67–) 2.19–2.45 (–3.02) μm thick; they were ellipsoidal from the side, reddish-brown when age, and with an L/W ratio = 1.2 ± 0.1. *Hyphae* hyaline, septate, smooth-walled, and (1.11–) 1.71–2.16 (–3.00) μm wide. *Chlamydospores* in PDA were hyaline, globose, clavate, doliiform, multi-guttulate, and were (5.37–) 9.19–10.59 (–14.1) μm × (5.69–) 7.42–8.30 (–10.28) μm, with an L/W ratio = 1.3 ± 0.3. *Conidia* in Leonian's agar were sparse, hyaline, non-septate, cylindrical to ellipsoidal, occasionally fusiform, occasionally guttulate, and (4.1–) 4.79–5.23 (–6.67) μm × (0.93–) 1.37–1.52 (–2.31) μm wide, with an L/W ratio = 3.5 ± 0.7; microcyclic conidiation was also observed.

Material examined: CHILE, Pinto, Ñuble, 36.818367 S, 71.622217 W; altitude: 697 m.a.s.l. endophyte strain of *Fragaria chiloensis* subsp. *chiloensis* f. *patagonica*, 13 July 2020. Collected by J.F. Castro and M. Guerra, isolated by Cecilia Santelices. Deposited in the Chilean Collection of Microbial Genetic Resources (CChRGM) with the code RGM 3311 and in the CBS under the code CBS 149292.

Notes: *Coniochaeta hansenii* RGM 3311 shared identical ITS and LSU sequences with *C. hansenii* CBS 885.68 (ITS identities: 183/183 (100%), no gaps, LSU identities: 456/456 (100%), no gaps), followed by *C. montana* ARIZ-AZ0093 (ITS identities: 346/374 (90.1%), 4 gaps). In the ML phylogenetic tree, the strain was grouped with *C. hansenii* CBS 885.68 with high bootstrap support. The ascospores in *C. hansenii* RGM 3311 were discoidal and nearly 128-spored, which matches the description for the species *C. hansenii*, according to the taxonomic key of *Coniochaeta* species developed by Doveri [15]. In this work, we described the anamorph of *C. hansenii*. The strain *C. hansenii* RGM 3311 grew at a pH of 5 and 6 and in the presence of 1% NaCl. This strain oxidized the following Biolog GEN III substrates: acetic acid, *N*-acetyl-D-galactosamine, *N*-acetyl-β-D-mannosamine (weakly), *N*-acetyl-D-glucosamine, *N*-acetyl neuraminic acid, L-alanine, γ-amino-butryric acid, D-arabitol (weakly), D-aspartic acid, bromo-succinic acid, D-cellobiose, citric acid, dextrin, formic acid, D-fructose-6-PO$_4$, L-fucose, L-galactonic acid lactone, D-galacturonic acid, gelatin, gentiobiose, D-gluconic acid, α-D-glucose, D-glucose-6-PO$_4$, glucuronamide (weakly), D-glucuronic acid, L-glutamic acid, glycerol, glycyl-L-proline, L-histidine, α-hydroxy-butyric acid, β-hydroxy-D,L-butyric acid (weakly), inosine, α-keto-butyric acid, α-keto-glutaric acid, D-lactic acid methyl ester, L-lactic acid, β-D-lactose, D-malic acid, L-malic acid, D-maltose, D-mannose, D-melibiose, methyl pyruvate, β-methyl- D-glucoside, mucic acid, pectin, propionic acid, L-pyroglutamic acid, quinic acid, L-rhamnose, D-saccharic acid, D-salicin, D-serine, D-trehalose, D-turanose, and tween 40, but it does not oxidize sucrose, stachyose, D-raffinose, D-fructose, D-galactose, 3-methyl glucose, D-fucose, D-sorbitol, D-mannitol, myo-inositol, L-arginine, L-aspartic acid, L-serine, *p*-hydroxy-phenylacetic acid, and acetoacetic acid. It was sensitive to 4% NaCl, 8% NaCl, guanidine HCl, niaproof 4, lithium chloride, sodium butyrate, but not to aztreonam, fusidic acid, lincomycin, minocycline, nalidixic acid, rifamycin SV, troleandomycin, or vancomycin, nor to D-serine, sodium bromate, 1% sodium lactate, potassium tellurite, tetrazolium blue, or tetrazolium violet.

## 4. Conclusions

This work contributes to expanding our current knowledge regarding the geographic extent and host source of species of the genus *Coniochatea*. Here, we report the description of two endophytic *Coniochaeta* strains, isolated from *Fragaria chiloensis* subsp. *chiloensis* f. *patagonica* collected from the foothills of the Chilean Andes.

Phylogenetic studies and macro- and micro-morphological analyses confirmed that strain RGM 3311 corresponded to *C. hansenii*, which to the best of our knowledge corresponds to the first report of this strain in Chile. Additionally, we described the previously unknown anamorph of this species. The second isolate corresponded to a novel species of *Coniochaeta*, named *C. fragariicola* sp. nov. RGM 3301$^T$ = CBS 149284$^T$ that was closely related to *C. ambigua* PAD S00027$^T$, *C. cephalothecoides* L821, *C. endophytica* AEA 9094$^T$, and *C. simbalensis* NFCCI 4236$^T$. The conidial width of *C. fragariicola* sp. nov. RGM 3301$^T$ was a distinctive feature of this novel species.

It is worth mentioning that some reference species of the genus *Coniochaeta* have only been described in morphological terms, and the genetic barcoding scheme of other species of this genus is incomplete, making it difficult to assign names to new isolates. We suggest a revision of the genus *Coniochaeta*, including the sequencing of genetic markers from reference specimens with their incomplete genetic barcoding as well as the sequencing of other molecular markers. We also suggest including phenotypic analyses to account for another layer of information when analyzing the taxonomic position of new isolates. Further research is needed to understand the interaction between these endophyte species and the Chilean wild strawberry plant.

**Author Contributions:** Conceptualization, J.F.C. and C.S.; methodology, J.F.C., C.C.-Q., M.G., J.C.-F., Y.O., B.T. and J.O.-C.; software, J.F.C., C.C.-Q., M.G. and J.C.-F.; validation, J.F.C. and C.S.; investigation, D.C.-G., C.C.-Q., J.C.-F. and M.G.; resources, J.F.C. and L.B.-B.; data curation, J.F.C., C.C.-Q. and J.C.-F.; writing—original draft preparation, J.F.C.; writing—review and editing, J.F.C., C.S., B.T. and C.C.-Q.; visualization, J.F.C., C.C.-Q. and C.S.; supervision, J.F.C.; project administration, J.F.C.; funding acquisition, J.F.C. All authors have read and agreed to the published version of the manuscript.

**Funding:** This research was funded by the FONDECYT INICIACIÓN (Grant: 11191074) from the Chilean National Agency for Research and Development (ANID) and the FONDEQUIP Program of ANID (Grant: EQM200205).

**Data Availability Statement:** GenBank accession numbers submitted in this work are OP962067-70. Phylogenetic data are available from Dryad: https://datadryad.org/stash/share/QP9NZ9 6mvlEabPd4M7i2If8bW7oXx7JZnhgstznMIh4, accessed on 25 March 2023.

**Conflicts of Interest:** The authors declare no conflict of interest.

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
