# Peer review of "Description of Two Fungal Endophytes Isolated from Fragaria chiloensis subsp. chiloensis f. patagonica: Coniochaeta fragariicola sp. nov. and a New Record of Coniochaeta hansenii"

_2673-6500, doi:10.3390/taxonomy3020014_

Round 1
Reviewer 1 Report
Dear Authors
I read your manuscript entitled: Description of two fungal endophytes isolated from Fragaria chiloensis subsp. chiloensis f. patagonica from the Chilean Precordillera: Coniochaeta fragariicola sp. nov. and a new record of Coniochaeta hansenii (Manuscript ID: taxonomy-2135333) submitted to Taxonomy. Overall, your manuscript is well written and provides useful taxonomic information considering the introduction of a new species. Therefore, I recommend acceptance of this manuscript for publication after major revision. Regarding to this manuscript the following points should be considered. In addition, for all my comments and more details, please check the attached pdf file. Applying the requested changes and suggestions can be helpful in improving your manuscript for acceptance.
Good luck.
Some general issues:
1-Title:
The title seems a bit long and it is better to consider a shorter and comprehensive title for this manuscript.
2-Abstract:
In the Abstract section, there is confusion and repetition in some sentences. It is better for the authors to consider a proper framework for this section. For example, start from the host and then reach the materials and methods, results and summary discussion (The new information of this study compared to previous studies).
3- Introduction:
The introduction seems appropriate and satisfactory, but it needs to be revised in some parts (Some items are shown in the text).
4- Materials and Methods
This section is written almost completely and satisfactorily and only needs to change some words or sentences.
5- Results and discussion
It is better to organize the results and discussion based on the framework presented in the materials and methods. For example, 1- morphological features, 2- molecular identification, and 3- phylogeny and the results of their analysis. In addition, it may be better to use more references (if possible) in the discussion section.
6-Tables and Figures:
Some images and Tables need to be revised as shown in the pdf file.
7-References:
The references, especially the names of the journals, are not written uniformly and should be revised according to the guidelines of the journal (Taxonomy).

Author Response
Dear Reviewer,
We have submitted our replies in the attached file. We appreciate your time in reviewing our manuscript.
With best wishes,
Jean Franco Castro

Reviewer 2 Report
The authors present interesting new taxonomic data on parts of the genus Coniochaete.
Line 4. The authors provide their family names first and their given names last. This goes against the MDPI Taxonomy style.
13. “plant,” > “plant”
14 and 187. “the Coniochaete genus” > “the genus Coniochaete”
14. It makes sense to indicate where in the fungal tree of life Coniochaete belongs – for example by providing order and phylum upon first mention of the genus.
19. “Coniochaete fragariicola” is not an “epithet” but a name. Thus, I propose “epithet” > “name”.
23. “corresponds to” > “represents”
23 and elsewhere. When the authors say that they present “a new report” of C. hansenii in Chile, it means that C. hansenii has been found in Chile before, and now the authors found it again. But is that really what the authors want to say? My guess is that this species was found in Chile for the first time. If this is correct, “a new report” > “for the first time” or something like it. If not, then please provide information on the previous recoveries of this species in Chile.
26 and elsewhere. It’s not clear what the authors mean with “share more similarities with”. More than what? Please clarify, here and elsewhere.
27. “conidia width” > “conidial width” or “the width of the conidia”
33. The ICTF requests all mycologists to always write all fungal Latin names in italics also beyond the genus level (https://imafungus.biomedcentral.com/articles/10.1186/s43008-020-00048-6). The authors ignore this recommendation by writing the family and order name in regular typeface. Why?
34 and many places elsewhere. The authors variously use the Oxford comma, variously not. This gives a inconsistent impression. They use it on line 27 but not on this line, for instance. Please resolve this in a consistent way throughout the manuscript.
52. “have also” > “have”
55. I propose: “[12-14]. Other species are known from their sexual…”
92. I cannot find these two cultures in the CBS database. Are they withheld pending publication? Or is something wrong with their accession numbers?
110. I propose: “the dung was ground and resuspended”
112. “Crous,” > “Crous”
128. “The internal” > “The nuclear ribosomal internal”
132. “locus/loci” is a genetical term that cannot be used like this. I recommend “genetic markers” instead.
140. It is standard procedure to exercise at least basic quality control of newly generated DNA sequences by following, e.g., https://mycokeys.pensoft.net/articles.php?id=1186 . The authors, however, make no mention of having done quality control. This is a bit scary. We don’t need yet another mycological paper whose results turned out to be based on shaky sequence data.
147. “The phylogenetic” > “A phylogenetic”
150. Models of sequence evolution are normally inferred separately for ITS1, 5.8S, and ITS2 in order not to average away important information. The authors, however, lump them all into a single region and estimate a model for the full region. Why?
151. Please provide the version number of ModelTest and all other software tools used in the making of the study. In the interest of scientific reproducibility, obviously.
154. Nothing is said about tree rooting or outgroup choice. This is most unusual. Why was this information left out from the manuscript?
Table 1. It would be useful if the authors could highlight (through, e.g., boldface) all types/ex-type cultures in the table.
174. “for biomass production” > “for assessment of biomass production”
181-183. Nice to see that freely available software was used!
189 and elsewhere. “clustered” – however, since no clustering analysis was done, this should be “formed a clade with” instead.
191 “could be identical species” > “represent one and the same species”. This is what the authors meant, right?
199-203. This is not a complete sentence since a verb is missing. Pleas clarify this long and complicated sentence.
201. I propose: “. No LSU sequence was available. Our results suggest that Conio…”
207. Please clarify “they are a diverse clade”.
208. “strongly” > “closely”
217. What does it mean to “share” a multiple sequence alignment? Please clarify.
228. I propose: “but as a separate species. We propose the…”
230. The authors violate the MIAPA standard (https://pubmed.ncbi.nlm.nih.gov/16901231/) by withholding the multiple sequence alignment and phylogenetic tree files. These should be deposited in TreeBase or Dryad, or perhaps as supplementary items with the manuscript. They should furthermore be available for review.
It is common procedure when one describes a new species to do a BLAST search in GenBank to see if the species has been found before. In this case, it has. When I use the ITS2 of OP962067, you find that more than 10 identical ITS2 sequences have been found in other studies. This will allow the user to say something about the geographical distribution (and perhaps ecology) of the new species. The authors should probably also do an ITS2 BLAST search in UNITE (https://unite.ut.ee/) to see if the species has been recovered through metabarcoding.
Figure 2 is clever!
297 and many places elsewhere. “allantois” should presumably be “allantoid” – that is sausage-shaped. “allantois” is something else entirely (https://en.wikipedia.org/wiki/Allantois).
294. “as it occurs in” – please clarify.
300. “and additionally with formation of mucous heads … was not observed” – please clarify this complicated sentence.
305. “as” > “as does”
312. “develop the” > “develop any”
318. “condition” > “a condition”
322. Please clarify “as well as in”
329. “displayed” > “displayed a”
341. “does not” > “did not”
345 and 397. “otherwise specified” should presumably be “unless otherwise specified”. But this does not make sense. Are they authors saying that they are not really sure when they should provide this information and when they should withhold it?
357. “near 128” > “near 128 in number”
366. “such as,” > “such as”
371. “featuring a neck with short-conical, similar to” – a noun is missing here.
377. “in both,” > “in both”
380. “shorter” > “smaller”
384. “description” > “a description”
387. I propose: “that developed a teleomorph that contained”
393. “although” > “although they”
404 and on. “; the disaccharides …”. Verbs are generally missing in these incomplete and hard-to-read sentences. I think these sentences should be simplified. No need for ; or : in them. It is enough to use comma.
417. “utilized the” > “utilized”
422. “oxidized the” > “oxidized”
428. “vancomycin; as well as” – please simplify these enumerations. It is nice, though, that the authors took the time to do all of these tests!
523. “and, neither, …” – please simplify these enumerations.
531. Again “more similarity” – more than what?
535. It is domewhat ironic that the authors complain about the lack of DNA sequences for the genus Coniochaete, when clearly they have not looked at the DNA sequences in GenBank (BLAST) to see what relevant sequences are available. Like I said, more than 10 identical sequences are available for the new species as far as ITS2 goes.
566. The journal name is abbreviated, no dots are used.
585. The journal name is abbreviated, dots are used.
581. The journal name is not abbreviated.
Please specify the references in the way outlined in the author instructions.
668. “Interesting Fungi” > “interesting fungi”
Author Response

(The authors gave the same response as above.)

Reviewer 3 Report
It is a standard taxonomic paper with clear writing and logical flow. I'd recommend using the term consistently throughout. On some parts, the authors use 'sexual morph'. In some parts, I see 'teleomorph'. I'm not aware of any subtle differences between the two terms. If there is, please define in this paper.
The tree also looks really busy with support value on every node. I suggest removing support values that have SH-aLRT < 80 and UF-bootstrap < 95. It is unclear where some of the support values point at. Many nodes don't add anything to the story. Removing some of them will make the tree much cleaner. Also the authors chose to include several species twice (such as C. punctulata and C. taeniospora) or three times (C. pulveracea) without clear justification. This makes the tree longer than it needs for the sole purpose of describing 2 'new' species. Removing some samples might make a tree fit in 1 page and more readable.
Fig. 4b is unclear as the perithecium is partially blocked. What is the purpose of 1C when there are already better pictures of asci. Fig. 4l is also not of good quality, it could be zoomed in to see the chlamydospores.
Average size and Q-ratio should be given for conidia and chlamydospores of both species.
Lastly, based on BLAST results, are there any environmental sequences of Coniochaeta that are similar to these 2 species? This could give us a clue on what their ecological role might be.
Author Response

(The authors gave the same response as above.)

Round 2
Reviewer 1 Report
Dear authors
I read the revised version of the manuscript entitled: Description of two fungal endophytes isolated from Fragaria chiloensis subsp. chiloensis f. patagonica from the Chilean Precordillera: Coniochaeta fragariicola sp. nov. and a new record of Coniochaeta hansenii (taxonomy-2135333) submitted to Taxonomy. Thank you for making the requested corrections and applying the suggestions of the reviewers. I think that revised version of your manuscript is ready for acceptance.
Best regards.
Author Response
We thank the reviewer for revising our manuscript carefully.
Reviewer 2 Report
The authors return with a much improved manuscript. It is still the case that the manuscript should be gone through by a native English speaker, though.
15. ” Precordillera,” > ” Precordillera”
28. Oxford comma missing.
34. ” These group of” > ”This group of”
40. Oxford comma missing.
89. ”provider’s” > ”manufacturer’s” ?
88-93, 153-163, 329-345, 367-378, 427-430. No need for semicolons here. Why not simply write this text as regular sentences?
115. ” retrieved previous” > ”retrieved from previous”
194. ”, revealed” > ” revealed”
207. ” leads to infere” > ”lead us to conclude”
216. ”suggesting” > ”, suggesting”
225. ”clades, ” > ”clades”
239. ”were” > ”was”
274. ”whorl” > ”whorls” ?
304. ”were not” > ”was not”
324. ”whereas,” > ”whereas”
339. ”unlike” > ”unlike those of”
383. ” asci, developed by Doveri [15],” > ” asci developed by Doveri [15] indicates”
390. ”such as, ” > ”such as”
391. ”displayed” > ”had”
472, 527. ”and” > ”or”
504. ”clustered” – the authors said that they have removed all reference to clustering (since no clustering analysis was done), but I guess they didn’t after all?
533. ”studies, ” > ”studies”
535. ” we described the anamorph of this species, for the first time” > ”we describe the previously unknown anamorph of this species.”
543. ” making difficult the species assignation to new isolates” > ”which makes it difficult to assign names to new isolates”
541. The concept of ”reference species” is introduced very abruptly here. What does it mean? I think the authors have ”reference specimens” in mind.
547. ”Also, ” can be deleted.
Author Response
We thank the reviewer for the time invested in revising our MS. The current version has been proofread as required.